# Mendelian Randomization Study on Amino Acid Metabolism Suggests Tyrosine as Causal Trait for Type 2 Diabetes

**DOI:** 10.3390/nu12123890

**Published:** 2020-12-19

**Authors:** Susanne Jäger, Rafael Cuadrat, Clemens Wittenbecher, Anna Floegel, Per Hoffmann, Cornelia Prehn, Jerzy Adamski, Tobias Pischon, Matthias B. Schulze

**Affiliations:** 1Department of Molecular Epidemiology, German Institute of Human Nutrition Potsdam-Rehbruecke, 14558 Nuthetal, Germany; Rafael.Cuadrat@dife.de (R.C.); cwittenbecher@hsph.harvard.edu (C.W.); mschulze@dife.de (M.B.S.); 2German Center for Diabetes Research (DZD), 85764 Neuherberg, Germany; adamski@helmholtz-muenchen.de; 3Department of Nutrition, Harvard T.H. Chan School of Public Health, Boston, MA 02115, USA; 4Leibniz Institute for Prevention Research and Epidemiology-BIPS, 28359 Bremen, Germany; floegel@leibniz-bips.de; 5Human Genomics Research Group, Department of Biomedicine, University of Basel, 4031 Basel, Switzerland; per.hoffmann@unibas.ch; 6Institute of Human Genetics, Division of Genomics, Life & Brain Research Centre, University Hospital of Bonn, 53105 Bonn, Germany; 7Research Unit Molecular Endocrinology and Metabolism, Helmholtz Zentrum München, German Research Center for Environmental Health, 85764 Neuherberg, Germany; prehn@helmholtz-muenchen.de; 8Chair of Experimental Genetics, Center of Life and Food Sciences Weihenstephan, Technische Universität München, 85354 Freising-Weihenstephan, Germany; 9Department of Biochemistry, Yong Loo Lin School of Medicine, National University of Singapore, 8 Medical Drive, Singapore 117597, Singapore; 10Molecular Epidemiology Research Group, Max Delbrueck Center for Molecular Medicine in the Helmholtz Association (MDC), 13125 Berlin, Germany; tobias.pischon@mdc-berlin.de; 11Charité–Universitätsmedizin Berlin, Corporate Member of Freie Universität Berlin, Humboldt-Universität zu Berlin and Berlin Institute of Health (BIH), 10117 Berlin, Germany; 12MDC/BIH Biobank, Max Delbrueck Center for Molecular Medicine in the Helmholtz Association (MDC) and Berlin Institute of Health (BIH), 13125 Berlin, Germany; 13Institute of Nutritional Science, University of Potsdam, 14558 Nuthetal, Germany

**Keywords:** Mendelian randomization, amino acids, tyrosine, type 2 diabetes, GWAS

## Abstract

Circulating levels of branched-chain amino acids, glycine, or aromatic amino acids have been associated with risk of type 2 diabetes. However, whether those associations reflect causal relationships or are rather driven by early processes of disease development is unclear. We selected diabetes-related amino acid ratios based on metabolic network structures and investigated causal effects of these ratios and single amino acids on the risk of type 2 diabetes in two-sample Mendelian randomization studies. Selection of genetic instruments for amino acid traits relied on genome-wide association studies in a representative sub-cohort (up to 2265 participants) of the European Prospective Investigation into Cancer and Nutrition (EPIC)-Potsdam Study and public data from genome-wide association studies on single amino acids. For the selected instruments, outcome associations were drawn from the DIAGRAM (DIAbetes Genetics Replication And Meta-analysis, 74,124 cases and 824,006 controls) consortium. Mendelian randomization results indicate an inverse association for a per standard deviation increase in ln-transformed tyrosine/methionine ratio with type 2 diabetes (OR = 0.87 (0.81–0.93)). Multivariable Mendelian randomization revealed inverse association for higher log_10_-transformed tyrosine levels with type 2 diabetes (OR = 0.19 (0.04–0.88)), independent of other amino acids. Tyrosine might be a causal trait for type 2 diabetes independent of other diabetes-associated amino acids.

## 1. Introduction

Blood levels of amino acids have been associated with risk of type 2 diabetes or insulin resistance in several observational studies [1]. However, it is still not fully clear whether these metabolites are causal markers for disease etiology or rather reflect early pathophysiological processes of the disease development.

To investigate causal relationships between risk factors and disease outcomes, genetic variants can be used as instrumental variables in Mendelian randomization (MR). The principle is based on Mendels’s second law of inheritance, stating that the assignment of alleles is random during meiosis. Therefore, genetic variants are generally not associated with possible confounders (e.g., environmental exposures or lifestyle factors) in exposure-disease associations [2]. Furthermore, the genetic constitution of an individual is determined prior to the disease onset overcoming the chance of reverse causation [2]. Various previous MR studies investigated single amino acids in relation to type 2 diabetes risk but found controversial results. For glycine, one study found a causal inverse association with type 2 diabetes [3]; however, two others did not [4,5] and one suggested glycine levels rather being causally influenced by insulin resistance [5]. Phenylalanine showed a positive causal effect on risk of type 2 diabetes [3]. While branched-chain amino acids (BCAA) (isoleucine, leucine, and valine) have been associated with risk of type 2 diabetes using MR [6], others found no indication for a causal effect on insulin resistance, but rather the opposite direction of insulin resistance having a causal impact on BCAA levels [7].

Previous MR studies used large data sources; however, they focused on single amino acids only and did not account for their interrelations within human metabolism. Amino acid ratios which account for intercorrelations between single amino acids and utilizing underlying network structures might depict pathway reactions connected to functional SNPs [8]. Another novel approach to consider pleiotropic effects within complex metabolic pathways is multivariable MR, utilizing multiple genetic instruments to disentangle direct causal effects of risk factors [9].

Within this project, we aimed to use human genetics to estimate causal effects of single diabetes-related amino acids and amino acid ratios, as identified in the European Prospective Investigation into Cancer and Nutrition (EPIC)-Potsdam study (Appendix A) [10], on the risk of type 2 diabetes and to study which of the diabetes-related amino acid traits predominantly account for the causal relationship with risk of type 2 diabetes.

## 2. Materials and Methods

### 2.1. Workflow

Figure 1 illustrates the workflow of our analysis. First, we selected diabetes-related amino acid traits based on metabolic network structures within EPIC-Potsdam (Appendix A). Second, we identified suitable genetic instruments in genome-wide association studies (GWAS) on diabetes-related amino acid traits within EPIC-Potsdam data (*n* = 2265). Third, we evaluated the pathway enrichment of the genetic instruments to facilitate biological interpretation facilitating biological interpretation. Fourth, for the selected SNPs, disease associations were extracted from large public GWAS on type 2 diabetes and we performed univariable two-sample MR studies to estimate the causal effects of amino acid traits on type 2 diabetes risk (DIAGRAM data). To complement our results achieved within EPIC-Potsdam data and to gain higher power to generate genetic instruments for single amino acids, we used previously published GWAS data on single amino acid measures [11]. Fifth, these public data were furthermore used in a multivariable MR to estimate direct causal effects of diabetes-related amino acid traits. Finally, we evaluated the role of pre-existing insulin resistance (defined by fasting insulin in MAGIC data) by applying a reverse MR approach in EPIC-Potsdam and utilizing public GWAS data. Detailed description of data sources follows under the study populations section.

### 2.2. Study Populations

#### 2.2.1. Individual-Level Data from the EPIC-Potsdam Study

The European Prospective Investigation into Cancer and Nutrition (EPIC)-Potsdam study consists of 27,548 participants recruited between 1994 and 1998 from the general population in Potsdam and surroundings. The baseline examination involved a personal interview including questions on prevalent diseases, self-administered questionnaires, interviewer-conducted anthropometric measurements, and a blood sample collection [12]. We used a random sample within the EPIC-Potsdam study, described in detail previously [13]. Briefly, a sub-cohort of 2500 individuals was randomly selected from 26,444 participants who provided blood samples at baseline. Of these, participants with prevalent diabetes or diabetes medication at baseline were excluded. Further exclusion criteria were missing genetic data (described in detail within the genotyping and quality control section) and missing or implausible data on amino acid measurements, leaving up to 2265 individuals for analyses in the sub-cohort (Appendix A).

All participants provided written informed consent. The EPIC-Potsdam study was approved by the ethics committee of the State of Brandenburg, Germany (approval code: AS 29/93). All procedures were in accordance with the ethical standards of the institutional and/or national research committee and with the 1964 Helsinki declaration and its later amendments or comparable ethical standards.

#### 2.2.2. Summary-Level Data for Single Amino Acids

Estimates for the associations between genetic instruments and single amino acids (glycine, phenylalanine, tryptophan, tyrosine, methionine, valine, leucine, serine) were extracted from public GWAS data by Shin et al. investigating up to 7824 adult individuals with European ancestry from the KORA (Kooperative Gesundheitsforschung in der Region Augsburg) and the TwinsUK studies [11]. Amino acid traits are log_10_-transformed and beta estimates are adjusted for age and sex. We selected this data source, as all investigated amino acid traits from EPIC-Potsdam are available.

#### 2.2.3. Summary-Level Data for Type 2 Diabetes

Estimates for the association of amino acid-associated SNPs with type 2 diabetes were obtained from the DIAbetes Genetics Replication And Meta-analysis (DIAGRAM) consortium, including 32 studies with a total of 898,130 individuals (74,124 with type 2 diabetes and 824,006 without) of European ancestry [14]. We used the type 2 diabetes-risk estimates without BMI adjustment. Type 2 diabetes was defined by study-specific diagnostic criteria, ranging from self-reports, diabetes medication to verified diagnoses (see Appendix A) [14].

#### 2.2.4. Summary-Level Data for Fasting Insulin

Estimates for the associations between genetic instruments and fasting insulin were obtained from MAGIC (the Meta-Analyses of Glucose and Insulin-related traits Consortium) investigators [15]. The estimates are adjusted for age and sex and were generated from up to 108,557 individuals with European ancestry from 56 studies.

### 2.3. DNA-Extraction, Genotyping and Quality Control within EPIC-Potsdam

The DNA was extracted from buffy coats using the chemagic DNA Buffy Coat Kit on a Chemagic Magnetic Separation Module I (PerkinElmer chemagen Technologies, Baesweiler, Germany) according to the manufacturer’s manual. Samples from the EPIC-Potsdam participants were genotyped with three different genotyping arrays: Human660W-Quad_v1_A (*n* = 363), HumanCoreExome-12v1-0_B (*n* = 632, two datasets) and Illumina InfiniumOmniExpressExome-8v1-3_A DNA Analysis BeadChip (*n* = 1369). Genotyping and quality control of the Human660W-Quad_v1_A and HumanCoreExome-12v1-0_B chips was described elsewhere [16]. Genotyping using the Illumina InfiniumOmniExpressExome-8v1-3_A DNA Analysis BeadChip was performed in the Life and Brain Center in Bonn, Germany. Detailed description of genotyping and quality control and imputation was previously published [17]. Briefly, pre-/phasing and imputation were conducted using Eagle2 [18] and the Michigan Imputation Service [19] with The Haplotype Reference Consortium (release 1.1) as reference panel [20]. Imputation was conducted in four separate datasets (one for each chip or two for the HumanCoreExome-12v1-0_B chip) using minimac3 [19]. Imputed files were merged, keeping the minimal R ^2^ score from the four files. SNPs were filtered by R^2^ keeping those with values bigger than 0.6. Overall, we had genotype data for *n* = 2303 samples from the sub-cohort available (Appendix A).

### 2.4. Metabolite Measurements in EPIC-Potsdam

Metabolite quantification was performed in the Genome Analysis Center at the Helmholtz Zentrum München. All samples have been measured using the Absolute*IDQ*^TM^ p150 Kit (Biocrates Life Sciences AG, Innsbruck, Austria) in combination with flow injection analysis-tandem mass spectrometry (FIA-MS/MS). Serum samples of 10 µL serum were used to quantify 163 simultaneously, including 41 acylcarnitines (Cx:y), 14 amino acids, hexose (sum of six-carbon monosaccharides without distinction of isomers), 92 glycerophospholipids (lyso-, diacyl-, and acyl-alkylphosphatidylcholines), and 15 sphingomyelins. The method for sample preparation and measurement as well as the metabolite denomination was previously described [21]. All samples of EPIC-Potsdam have been processed in one batch [22]. We selected eight amino acid ratios out of the six diabetes-associated amino acids [10] which were connected via one edge [8] based on a network of amino acids (Appendix A) and two single amino acids, resulting in ten outcomes referred to as diabetes-related amino acid traits (glycine, phenylalanine, glycine/serine, serine/phenylalanine, phenylalanine/arginine, valine/xleucine, xleucine/Methionine, tyrosine/methionine, tyrosine/tryptophan, tryptophan/glutamine).

### 2.5. Statistical Analysis

Statistical Analysis System (SAS) Enterprise Guide 7.1 with SAS version 9.4 (SAS Institute Inc., Cary, NC, USA) was used for data management and preparation. QCtool v1.4 was used for genetic data filtering. SNPtest v2.5.2 [23] was used for GWAS. For Mendelian Randomization analyses we used R (version 3.6.3 (2020-02-29)) and the following R packages: TwoSampleMR (v0.5.4) [24], Mendelian Randomization (v0.4.2) [25], Radial MR (0.4) [26] and MVMR (v0.2) [27].

#### 2.5.1. Genome-Wide Association Study in EPIC-Potsdam

SNPs were filtered by SNP missing-rate (removed ≥ 0.05), minimum allele frequency (MAF) (removed out of interval (0.05–0.5)) and Hardy-Weinberg equilibrium (removed −log_10_(*p*-value) ≥ 3). We performed exploratory single variant association analysis using n ~ 5,338,500 markers as exposures, separately for each of the ten amino acid traits. The effective number of independent metabolite traits of 9 out of 10 was determined using equation 5 of Li and Ji [28]. We considered a *p*-value as genome-wide significant at *p* < 1.11 × 10^−8^ = [10^−7^/9]. Suggestive significance threshold was defined as *p* < 1.11 × 10^−6^ = [10^−5^/9]. Amino acid traits were natural log-transformed to normalize the right-skewed distributions. After ln-transformation, metabolite outliers of >4 standard deviations (SD) from the mean were excluded and amino acid traits were standardized (mean = 0; SD = 1). We assumed an additive genetic model, adjusted for age at recruitment and sex. Variants were mapped to Ensembl annotation version 87 (GRCh37) [29] and annotated by the Ensembl Variant Effect Predictor [30].

#### 2.5.2. Gene Set Enrichment Analysis Using EPIC-Potsdam GWAS Data

We used GSA-SNP2 software for gene set enrichment analysis based on GWAS *p*-Values [31]. This tool employs the *Z*-statistic of the random set model. The SNP to gene annotation was done using a 20 kilobases window upstream and downstream of the gene and highly correlated adjacent genes on European population were removed. We present pathways up to *q*-value <0.25 using the MSigDB C2.CP (curated canonical pathways) version 5.2 database [32,33,34] for pathway annotation that were grouped according to gold standard pathways for type 2 diabetes [31].

#### 2.5.3. Two-Sample Mendelian Randomization Analyses Using EPIC-Potsdam GWAS Data on Amino Acids and Ratios

We conducted a univariable two-sample MR study using amino acid traits as exposures on type 2 diabetes. Effect estimates of the association between genetic instruments and amino acid traits were obtained from EPIC-Potsdam data and effect estimates of the SNP-type 2 diabetes association were used from public summary GWAS data [14]. We selected suggestive significant (*p* < 1.11 × 10^−6^) or genome-wide significant (*p* < 1.11 × 10^−8^) instrumental variables and performed clumping according to linkage disequilibrium (LD). Therefore, SNPs within a window of 10,000 kb and being in LD as defined by R^2^ ≥ 0.3 were removed. The SNP with the lowest *p*-value was retained. Within the MR analysis, we accounted for their correlation among each other, estimated in 502 European samples from 1000 Genomes phase 3 as implemented within the Mendelian Randomization R package [25] (Appendix A). In sensitivity analysis, we performed strict clumping using the default threshold of R^2^ ≥ 0.001 from the TwoSampleMR R package to retain independent instruments. Data were harmonized for the direction of effects between exposure and outcome associations and palindromic SNPs were excluded. We then used an inverse variance weighted (IVW) meta-analysis of each SNP specific Wald ratio (SNP-outcome estimate divided by SNP-exposure estimate) using random effects or fixed effects (≤ 3 instruments), to obtain an estimate for the causal effect of the respective amino acid trait on type 2 diabetes. To investigate violations of the MR assumptions due to horizontal pleiotropy, we evaluated the MR-Egger intercept [35]. Heterogeneity was assessed by the Cochran’s Q statistic and if applicable we performed RadialMR using second order weights and an α level of 0.05 [26] to identify outliers with the largest contribution to Q statistic. Furthermore, we obtained the mean F statistic for the set of genetic variants as indicator of instrument strength [26].

#### 2.5.4. Two-Sample Mendelian Randomization Analyses Using Public GWAS Data on Single Amino Acids

To complement our results obtained from EPIC-Potsdam data and to gain higher power for generating genetic instruments for single amino acids, we used previously published GWAS data [11] and performed univariable two-sample MRs for single amino acids and type 2 diabetes. Genetic instruments were selected based on strict genome-wide significance threshold of 5 × 10^−8^ and default clumping threshold of R^2^ ≥ 0.001 [11].

Furthermore, we conducted multivariable MR using tyrosine, methionine, and tryptophan as exposures [11] on risk of type 2 diabetes [14] to receive direct causal effects for tyrosine adjusted for the direct neighbors within the amino acid network. Genetic instruments for each exposure were selected based on strict genome-wide significance threshold of 5 × 10^−8^ and default clumping threshold of R^2^ ≥ 0.001 [11] (Appendix A). Then we obtained all SNP effects on the respective other amino acids. Extraction of the outcome GWAS results and data harmonization was conducted as described before.

This analysis was extended by adjusting tyrosine by all other diabetes-related amino acids. We calculated F-statistics to evaluate the presence of weak instruments within the multivariable MR analysis and adjusted for those by minimizing the Q-statistic allowing for heterogeneity using “qhet_mvmr” function from the MVMR package [27]. Phenotypic correlation was obtained from EPIC-Potsdam data, considering levels of xLeu as being the sum of isoleucine and leucine, which were not directly measured in our study.

To account for horizontal pleiotropic effects by other pathways besides amino acids and to avoid false positive finding, we used Causal Analysis Using Summary Effect estimates (CAUSE) method [36] for tyrosine using three different data sources: Shin et al. [11], Kettunen et al. [37] and Locke et al. [38]. The latter two were also used for univariable MR with type 2 diabetes.

Furthermore, we tested for associations of tyrosine with type 2 diabetes-related outcomes like fasting blood glucose, fasting insulin, BMI, and waist circumference in univariable MR approach.

#### 2.5.5. Reverse Two-Sample Mendelian Randomization Analyses of Insulin Resistance and Amino Acids

We performed reverse MR analysis to test causality of insulin resistance on amino acid traits [5,7] using nine out of ten genetic instruments from a genetic risk score for fasting insulin as a marker of insulin resistance [39], providing a mean F statistic of 27.9. Results of amino acid ratios showing causal influence by insulin resistance in EPIC-Potsdam were complemented by using EPIC-Potsdam and public GWAS results for single amino acids [11].

## 3. Results

### 3.1. Selection of Genetic Instruments by GWAS on Amino Acid Traits in EPIC-Potsdam

GWAS were conducted within the EPIC-Potsdam study with up to 2265 participants. Baseline characteristics are illustrated in Table 1.

We identified genome-wide significant hits for glycine, glycine/serine, and tyrosine/methionine ratios. While hits such as *CPS1* for glycine and glycine/serine ratio or *PHGDH* for glycine/serine or serine/phenylalanine ratios were known from previous GWAS, we also identified novel suggestive associations such as *MEG9* for glycine and *GOT2* for tyrosine/methionine ratio. All GWAS hits are summarized in Table 2 and Appendix A. For tyrosine/tryptophan ratio, we did not identify suggestive associations (Appendix A).

### 3.2. Enrichment of Amino Acid-Associated SNPs in Type 2 Diabetes-Related Pathways

We analyzed which pathways were enriched with genetic signals on diabetes-related amino acids. Therefore, we used curated sets of type 2 diabetes-related pathways (Figure 2).

Genetic signals on tyrosine/methionine showed the most enriched pathways which were mainly related to NOTCH signaling. While phenylalanine/arginine was enriched in pathways mapping to fatty acid metabolism; glycine/serine, phenylalanine, and tryptophan/glutamine were mainly related to pathways of glucose metabolism and type 2 diabetes. GWAS results for glycine, serine/phenylalanine, tyrosine/tryptophan, xLeucine/methionine, and valine/xLeucine were not enriched in type 2 diabetes-related pathways.

### 3.3. Causal Estimates for Amino Acid Traits on Risk of Type 2 Diabetes

There was an inverse association with type 2 diabetes risk per SD increase in ln-transformed tyrosine/methionine ratio (beta = −0.141; OR = 0.87 (0.81–0.93)) when including one genome-wide significant instrument located in *SLC16A10* (Table 3, Appendix A).

Although heterogeneity was identified for suggestive hits of glycine/serine, tyrosine/methionine and xLeucine/methionine, outliers could only be detected for glycine/serine (rs2010825). Exclusion of this variant did not change the overall not significant association (Table 3). If applicable, we could not find indication for directional horizontal pleiotropy as assessed with MR-Egger regression (Table 3). Using public GWAS data on single log_10_-transformed amino acid measurements, we found that the association of tyrosine/methionine ratio seems to be driven by tyrosine showing itself a strong inverse association (beta = −1.530; OR = 0.22 (0.07–0.67)) (Figure 3, Appendix A). Higher levels of methionine, however, showed no significant association with type 2 diabetes, although the estimate was also quite strong (beta = −1.170; OR = 0.31 (0.04–2.2)) (Figure 3, Appendix A).

We investigated other diabetes-related amino acid traits using public data and found positive associations for increased levels of valine and leucine in univariable MR (Appendix A). When we adjusted tyrosine for its direct neighbors in the network (tryptophan and methionine) in multivariable MR, we found an inverse association for higher levels of tyrosine (OR = 0.27 (0.07–1.12), Figure 3), which became stronger and significant after adjustment for all other diabetes-associated amino acids (tryptophan, leucine, isoleucine, valine, glycine, and phenylalanine) (beta = −1.66; OR = 0.19 (0.04–0.88), Figure 2 and Appendix A) while neither higher levels of BCAA nor glycine, phenylalanine or tryptophan showed significant associations. F-statistics were below 10 for all exposures suggesting potentially weak instruments used in this analysis (Appendix A). However, we repeated the multivariable MR by accounting for weak instruments and received comparable effect estimate for tyrosine (beta = −1.76; OR = 0.17 (0.03–0.98), Appendix A).

Using CAUSE method for tyrosine, the model fit for both, the sharing and causal models, was not significantly different from the null model (Table 4) using data from Shin et al. or Kettunen et al. Using data from Shin et al., the causal model seemed to have a better fit than the sharing model; still, none of the models was statistically better than the other one. However, we could use only 83,936 genetic variants to estimate the nuisance parameters which might have resulted in bad model comparisons as it is advised to use at least 100,000 variants. When we used most recent data from Locke et al. [38], we found that the causal model was significantly better than the null or the sharing model (Table 4). However, in univariable MR using this data source, we found indication for significant positive association with type 2 diabetes, contrasting the effect directions observed from Shin et al. and Kettunen et al. (Appendix A).

Tyrosine showed no significant associations with BMI, waist circumference and fasting insulin, but results tended towards an inverse association with fasting blood glucose adjusted for BMI (*p* = 0.008) and blood glucose (*p* = 0.004). (Appendix A).

### 3.4. Causal Estimates for Insulin Resistance on Amino Acid Traits

We found significant associations of insulin resistance on serine/phenylalanine (beta = −1.895; OR = 0.15 (0.03–0.81)) and tyrosine/methionine (beta = 1.486; OR = 4.42 (1.03–19.0)) ratios in reverse MR analysis (Appendix A–S13) using EPIC-Potsdam data. The former was driven by the significant association of insulin resistance on serine; which was however only identified when using, EPIC-Potsdam data (beta = −1.75; OR = 0.17 (0.04–0.76), Appendix A). Insulin resistance was not causally linked to individual amino acid measures of phenylalanine, tyrosine, and methionine, neither using data from EPIC-Potsdam nor from Shin et al. [11] (Appendix A).

## 4. Discussion

Within this study, we applied a two-sample MR approach to investigate causal effects of diabetes-associated amino acid traits on the risk of type 2 diabetes. As a main finding, our MR analysis supports a causal effect of tyrosine on type 2 diabetes which seems to be independent of other diabetes-associated amino acids. For BCAA, glycine, tryptophan and phenylalanine, no direct causal effects could be established in this multivariable MR analysis. Reverse MR approach indicated causal effect of insulin resistance on tyrosine/methionine and serine/phenylalanine ratios in EPIC-Potsdam, but this could not be attributed to single amino acids within the ratios.

### 4.1. Amino Acids and Type 2 Diabetes

We found a genetically inverse association of tyrosine/methionine ratio with type 2 diabetes which seems to be driven by tyrosine. This inverse association was robust throughout all our analyses, independent of whether we used the tyrosine/methionine ratio in EPIC-Potsdam or tyrosine as single trait obtained from Shin et al. [11]. Furthermore, this inverse association was robust to adjustment by other diabetes-related amino acids in multivariable MR. Genome-wide significant instruments for tyrosine selected in public data were located on the same chromosomes as suggestive instruments identified in EPIC-Potsdam GWAS for tyrosine/methionine ratio; however, there was no indication for LD between the respective SNPs. Although, our results are in accordance with a recent MR study on tyrosine [41]; it is noteworthy that our findings (except the univariable MR using data from Locke et al. [38]) are in contrast to observational studies conducted in Caucasian populations (including EPIC-Potsdam) which found positive associations between serum tyrosine levels and type 2 diabetes risk [10,42].

With regard to the other investigated diabetes-related amino acid traits, utilizing ratios did not result in novel findings compared to previous MR studies on single amino acids. However, our multivariable approach combining all diabetes-associated amino acids into one model yielded no direct effect for BCAA, glycine, phenylalanine, or tryptophan. This is in contrast to our univariable MR on valine and leucine and previous univariable MRs on BCAA [6] and phenylalanine [3] showing risk increasing effects on type 2 diabetes. Previous MR studies might have come to divergent conclusions, as they have missed the intercorrelation between different amino acids in human metabolism by focussing on single traits only.

The association of tyrosine on type 2 diabetes could not be attributed to pre-existing insulin resistance. Although we found a significant effect of insulin resistance on tyrosine/methionine ratio in reverse MR approach in EPIC-Potsdam, we could not verify this result for single amino acids included within the ratio, neither in EPIC-Potsdam nor in larger GWAS [11]. Therefore, our results suggest that the direct association for tyrosine is not confounded by reverse causation due to insulin resistance. Still, this finding should be confirmed in future studies.

### 4.2. Biological Mechanisms Linking Tyrosine to Type 2 Diabetes

Our main genetic instrument for tyrosine was located within the *SLC16A10* locus. The locus was previously discovered to be associated with isoleucine/tyrosine ratio [43] (rs7760535, LD with our EPIC-Potsdam variant (rs17606481): r^2^ = 0.25 and D’ = 0.93). *SLC16A10* encodes a monocarboxylate transporter 10 also known as T-type amino acid transporter 1 (TAT1) which is a sodium-independent transporter that mediates the uptake of aromatic acids like tyrosine, tryptophan phenylalanine and the non-proteogenic amino acid L-DOPA [44]. Tyrosine might act on type 2 diabetes risk via its crucial role in synthesis of the neurotransmitter dopamine and L-DOPA. Although, research regarding the effects of dopamine on glucose homeostasis and beta cell function is still at the beginning, in vitro and in vivo studies suggest that dopaminergic agents exert actions on the central nervous system (e.g., appetite control) as well as on the endocrine system (pituitary and pancreas) which modulate glucose and energy homeostasis [45]. In addition to being endogenously synthesized from phenylalanine, tyrosine is found in high-protein foods such as meat, fish, and cheese, but also in non-animal products like nuts, legumes, and whole-grains. Especially the latter are considered as beneficial for health and diets rich in those foods have been associated with reduced risk for type 2 diabetes [46]. Whether dietary supplementation of tyrosine is able to modulate glucose tolerance is studied within an ongoing clinical trial [47]. A recent study in rodents showed that beta cell function is affected by oral tyrosine administration and postulated dopamine and L-DOPA derived from nutritional tyrosine as anti-incretin candidates [48]. On the one hand, high blood levels of tyrosine may result in high L-DOPA and dopamine levels and therefore strengthen the anti-incretin effect. On the other hand, high tyrosine levels may reflect a reduced tyrosine hydroxylase (TH) activity resulting in low L-DOPA and dopamine levels and thereby weaken the anti-incretin effect. While the former would be supported by the observational finding of higher tyrosine levels being associated with higher diabetes risk, the latter would, however, be more in line with the causal inverse effect of tyrosine on blood glucose levels and type 2 diabetes risk, identified in our multivariable MR. A SNP (rs10770141) in the human *TH* gene resulting in reduced expression of TH was associated with lower type 2 diabetes risk [49], supporting our results. The TH expression and activity can furthermore be regulated by insulin to maintain an appropriate dopaminergic tone, suggesting that conditions with impaired insulin signaling are accompanied with a reduced TH expression [50], and hence high tyrosine levels. This might be one explanation why observational studies find higher tyrosine levels to be associated with higher risk. Another mechanism how tyrosine might relate to reduced type 2 diabetes risk is via its fermentation by gut microbiota to 4-Cresol, improving glucose homeostasis and β-cell function [51]. Still, the direction of effect obtained in our univariable MR depended on the respective study population and the resulting instruments used for the SNP-exposure association. Therefore, we cannot finally clarify whether the tyrosine-type 2 diabetes association is inverse or positive.

Gene-set enrichment analysis highlighted pathways connected to NOTCH signaling to be enriched in our data. NOTCH signaling is important for cell-cell signaling and plays a role in pancreas development. *NOTCH2* was previously described as a type 2 diabetes susceptibility locus [52]. Furthermore, NOTCH signaling is connected to beta cell insulin secretion [53].

### 4.3. Strengths and Limitations

One limitation of our analysis is the exclusion of prevalent diabetes cases in the EPIC-Potsdam sample which was used for instrument selection and might have weakened the exposure betas. However, compared to the overall study sample this number was small and we aimed to exclude bias from reverse causation due to the fact that prevalent cases might have different blood levels of amino acids [54]. When we used public GWAS data, we could observe a similar direction of association for tyrosine, and therefore this result seems to be robust. Furthermore, our findings are restricted to Europeans only, and can therefore not be generalized to other ethnicities.

A major concern is related to the presence of pleiotropic pathways as the single amino acids are connected within the metabolite network and therefore not independent of each other. Although we used ratios to address limitations by previous MR studies that did not account for correlations between specific amino acids, our MR analysis using estimates from EPIC-Potsdam might be underpowered (e.g., 59.3% power [55] to find the same causal effect for insulin resistance on glycine levels compared to the previous study [5] using the same genetic instruments). We addressed this disadvantage by complementing our results from EPIC-Potsdam with data from three times larger GWAS on single amino acids [11] (yielding a power of one for insulin resistance on glycine levels [55]). Still, we cannot rule out that we had insufficient power to detect causal effects for other amino acid traits beyond tyrosine. We adjusted for horizontal pleiotropy due to other amino acids, by applying novel multivariable MR framework; however, direct estimates might be biased if weak genetic instruments are used (indicated by F-statistic <10) [27], which was the case for all our exposures. Nevertheless, when we accounted for those in the analysis, we obtained comparable beta estimates and retained a significant direct effect for tyrosine. We tried to correct for correlated and uncorrelated horizontal pleiotropy using CAUSE MR method which did support a causal model for tyrosine and type 2 diabetes using most recent data for tyrosine measurements.

## 5. Conclusions

In conclusion, our results suggest that tyrosine might play a causal role for type 2 diabetes development, which seems to be independent of other diabetes-associated amino acids or pre-existing insulin resistance. However, interpretation should consider limitations like potentially weak and pleiotropic genetic instruments and the unclear effect direction which need to be addressed in future studies.

## Figures and Tables

**Figure 1 nutrients-12-03890-f001:**
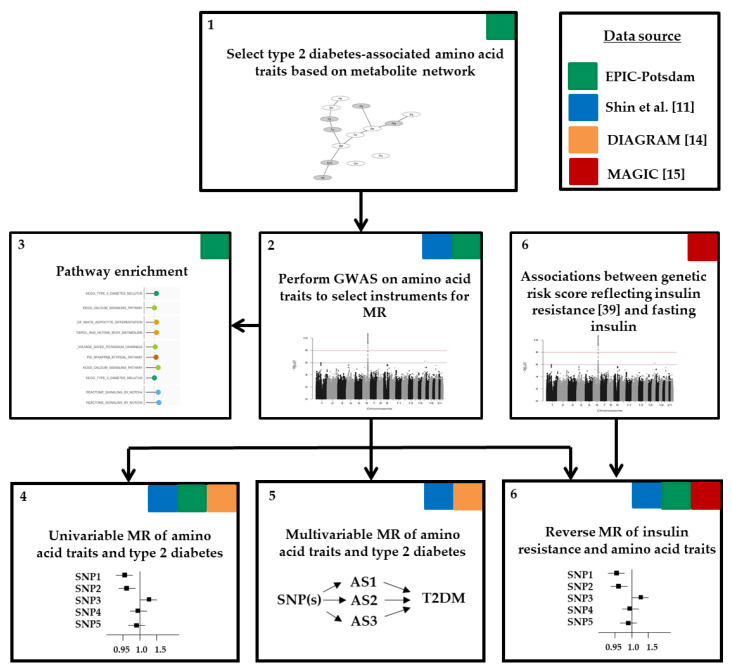
Workflow of analysis. AS, amino acid; DIAGRAM, DIAbetes Genetics Replication And Meta-analysis; GWAS, genome-wide association study; MAGIC, the Meta-Analyses of Glucose and Insulin-related traits Consortium; MR, Mendelian randomization; SNP, single nucleotide polymorphism; T2DM, type 2 diabetes mellitus.

**Figure 2 nutrients-12-03890-f002:**
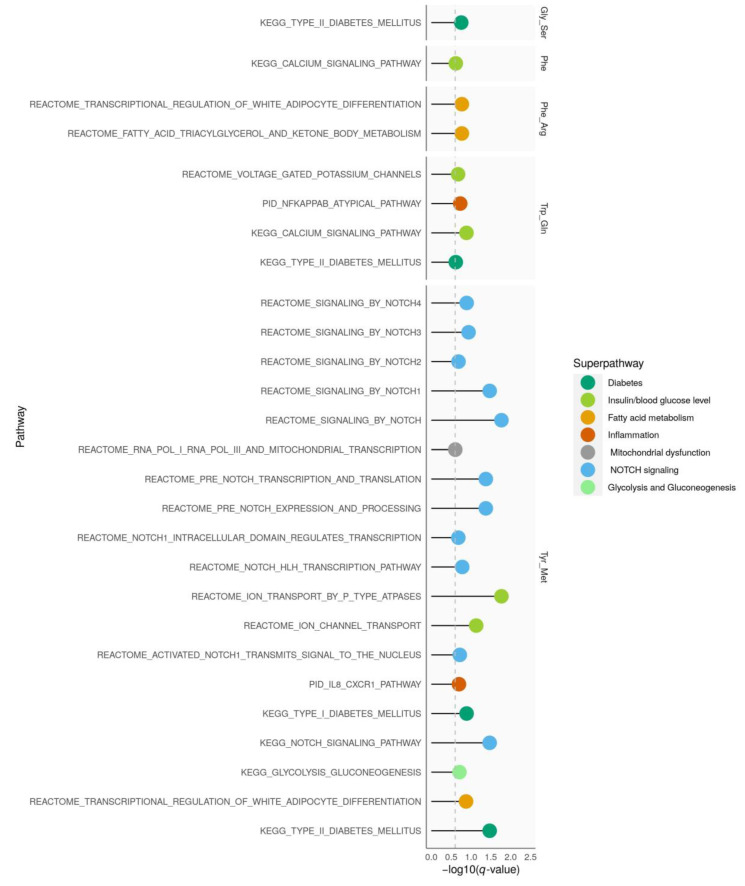
Pathway enrichment analysis. For each amino acid trait, pathways with significant enrichment of SNPs (*q*-value < 0.25) that mapped to gold standard pathways for type 2 diabetes are shown. Arg, arginine; Gln, glutamine; Gly, glycine; Met, methionine; Ser, serine; Phe, phenylalanine; Trp, tryptophan, Tyr, tyrosine.

**Figure 3 nutrients-12-03890-f003:**
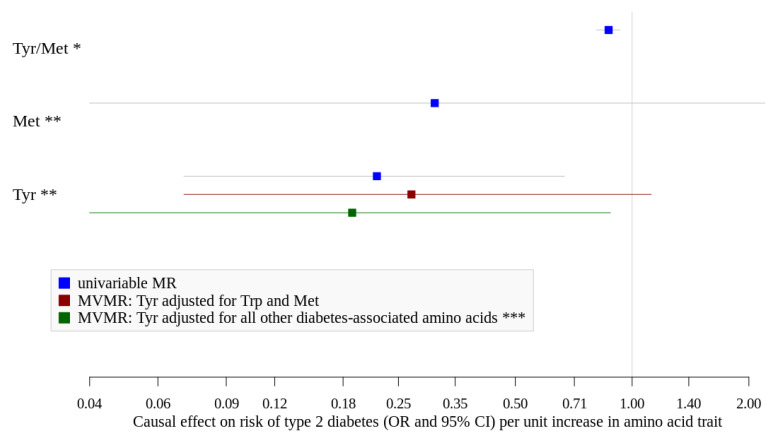
Causal effects (OR and 95%CI) of amino acid traits and type 2 diabetes. Met, methionine; Tyr, tyrosine. * assessed using EPIC-Potsdam GWAS data and Mahajan et al. [14], tyrosine/methionine ratio is ln-transformed and standardized. ** assessed using public GWAS data by Shin et al. [11] and Mahajan et al. [14], Met and Tyr are log_10_-transformed. *** adjusted for Tryptophan, Leucine, Isoleucine, Valine, Glycine, Phenylalanine.

**Table 1 nutrients-12-03890-t001:** Baseline characteristics, European Prospective Investigation into Cancer and Nutrition (EPIC)-Potsdam random sample.

	EPIC-Potsdam
*N*	2265
Sex (% men)	37.8
Age in years; median (interquartile range)	49.5 (15.6)
Waist circumference in cm; mean (SD)	85.3 (12.6)
Glycine in μmol/L; median (interquartile range)	241.0 (86.0)
Isoleucine + Leucine in μmol/L; median (interquartile range)	200.0 (70.0)
Phenylalanine in μmol/L; median (interquartile range)	55.0 (13.4)
Tryptophan in μmol/L; median (interquartile range)	80.0 (15.1)
Tyrosine in μmol/L; median (interquartile range)	78.7 (27.1)
Valine in μmol/L; median (interquartile range)	286.0 (83.0)

SD, standard deviation; Depicted are diabetes-associated amino acids.

**Table 2 nutrients-12-03890-t002:** Independent GWAS hits at suggestive significance *p* < 1.11 × 10^−6^ per amino acid trait in EPIC-Potsdam.

Amino Acid Trait	SNP	SNP Coordinates	Gene	EA/OA	EAF	N	Beta (SE)	*p*-Value	Consequence (GRCH37)	CADD-Score [40]
**Glycine/Serine**	rs1047891	2:211540507	*CPS1*	A/C	0.31	2265	0.50 (0.03)	7.58 × 10^−54^	Thr1406Asn	22.1
	rs2010825	7:44188220	*GCK*	C/T	0.51	2265	0.15 (0.03)	5.78 × 10^−7^	Downstream gene variant	5.97
	rs561931	1:120254506	*PHGDH*	G/A	0.61	2265	−0.18 (0.03)	2.28 × 10^−9^	Upstream gene variant	7.61
	rs9851577	3:125908310	*ALDH1L1; ALDH1L1-AS2*	T/C	0.46	2265	0.15 (0.03)	7.36 × 10^−7^	Intron variant	2.44
**Glycine**	rs1047891	2:211540507	*CPS1*	A/C	0.31	2264	0.59 (0.03)	2.79 × 10^−76^	Thr1406Asn	22.1
	rs61992673	14:101542105	*AL132709.1; MEG9*	A/C	0.20	2264	−0.22 (0.04)	1.62 × 10^−8^	Intron variant	5.34
**Phenylalanine/**	rs1451722	11:3856553	*RHOG*	T/C	0.58	2265	−0.15 (0.03)	3.51 × 10^−7^	Intron variant	12.7
**Arginine**	rs1718309	12:103242396	*PAH*	G/A	0.60	2265	−0.17 (0.03)	3.44 × 10^−8^	Intron variant	3.32
**Phenylalanine**	rs55940357	19:2610628	*GNG7; CTC-265F19.2*	C/T	0.88	2265	0.24 (0.05)	1.03 × 10^−6^	Intron variant	1.72
**Serine/**	rs1047891	2:211540507	*CPS1*	A/C	0.31	2265	0.18 (0.03)	5.95 × 10^−8^	Thr1406Asn	22.1
**Phenylalanine**	rs11024310	11:17520786	*USH1C*	G/C	0.08	2265	−0.31 (0.06)	9.83 × 10^−7^	Intron variant	1.25
	rs2992975	1:194106746	*-*	A/G	0.41	2265	0.17 (0.03)	5.65 × 10^−7^	Intergenic variant	0.63
	rs478093	1:120255126	*PHGDH*	G/A	0.69	2265	0.18 (0.03)	1.27 × 10^−8^	Intron variant	8.15
**Tryptophan/**	rs4903067	14:73286300	*DPF3*	C/T	0.34	2260	−0.15 (0.03)	6.28 × 10^−7^	Intron variant	0.88
**Glutamine**	rs7973936	12:64333645	*SRGAP1*	A/G	0.31	2260	0.18 (0.03)	3.07 × 10^−8^	Intron variant	1.58
**Tyrosine/**	rs12756904	1:104337030	*-*	C/T	0.20	2261	−0.18 (0.04)	1.08 × 10^−6^	Intergenic variant	2.96
**Methionine**	rs17606481	6:111542388	*SLC16A10*	G/A	0.15	2261	0.28 (0.04)	1.70 × 10^−11^	Intron variant	2.49
	rs72792419	16:58741949	*GOT2*	C/T	0.08	2261	−0.30 (0.06)	6.59 × 10^−7^	Downstream gene variant	4.23
**Valine/xLeucine**	rs2456586	19:51434353	*CTB-147C22.3*	C/T	0.39	2258	0.15 (0.03)	6.11 × 10^−7^	Upstream gene variant	0.64
**xLeucine/**	rs12642299	4:90942633	*-*	G/C	0.67	2263	0.17 (0.03)	6.16 × 10^−8^	Intergenic variant	0.99
**Methionine**	rs1958029	14:21491151	*NDRG2; AL161668.5; TPPP2*	G/A	0.11	2263	−0.25 (0.05)	1.84 × 10^−7^	Upstream gene variant	3.12

CADD, combined annotation dependent depletion; EA, effect allele; EAF, effect allele frequency; OA, other allele; SE, standard error. SNP, single nucleotide polymorphism.

**Table 3 nutrients-12-03890-t003:** Total causal effects of amino acid traits and of type 2 diabetes using instruments with R^2^ < 0.3.

Amino Acid Trait	Instruments	N (SNPs) ^a^	Beta (SE) from IVW	*p-*Value	Heterogeneity between SNPs; Q-Statistic, *p*-Value	Directional Horizontal Pleiotropy ^b^; Egger-Intercept (SE), *p*-Value	Outlier Detected
**Glycine**	suggestive	13/16	−0.003 (0.011)	0.804	no; 12.43,0.41	no; −0.001 (0.003), 0.85	no
	genome-wide	9/11	−0.005 (0.011)	0.686	no; 8.62, 0.38	no; 0.004 (0.004), 0.37	no
**Glycine/Serine**	suggestive	17/18	0.018 (0.019)	0.348	yes; 48.24, <0.001	yes; 0.009 (0.004), 0.02	Yes (rs2010825)
	suggestive (excluding outlier)	16/17	−0.001 (0.011)	0.895	no; 15.04, 0.45	no; 0.005 (0.003), 0.15	-
	genome-wide	12/12	−0.003 (0.012)	0.800	no; 10.47, 0.49	no; 0.006 (0.004), 0.14	no
**Phenylalanine**	suggestive	1/1	0.053 (0.045)	0.237	n.a.	n.a.	n.a.
**Phenylalanine/Arginine**	suggestive	2/2	0.043 (0.028)	0.120	no; 0.36, 0.54	n.a.	no
**Serine/Phenylalanine**	suggestive	3/4	0.000 (0.023)	0.990	no; 0.59, 0.74	no; 0.08 (0.26), 0.77	n.a.
**Tryptophan/Glutamine**	suggestive	2/2	0.002 (0.028)	0.931	no; 0.16, 0.69	n.a.	n.a.
**Tyrosine/Methionine**	suggestive	4/4	−0.012 (0.069)	0.857	yes; 29.59, <0.001	no; 0.04 (0.02), 0.08	n.a.
	genome-wide	1/1	−0.141 (0.033)	<0.001	n.a.	n.a.	n.a.
**xLeucine/Methionine**	suggestive	2/3	0.05 (0.031)	0.108	yes; 7.54, 0.01	n.a.	n.a.
**Valine/xLeucine**	suggestive	1/1	−0.012 (0.044)	0.777	n.a.	n.a.	n.a.

IVW, inverse variance weighted method; n.a., not applicable; SE, standard error. xLeucine = Isoleucine + Leucine. ^a^ used instruments/suitable instruments available in the GWAS of type 2 diabetes in Mahajan et al. [14] ^b^ assessed by MR-Egger (>2 variants needed).

**Table 4 nutrients-12-03890-t004:** Results from CAUSE MR for tyrosine with type 2 diabetes.

Data Source	*N* Variants Used to Calculate Nuisance Parameters	*N* Variants to Estimate CAUSE Posteriors	Model 1 *	Model 2 *	∆ ELPD **	SE ∆ ELPD	*z*-Score	*p*-Value
**Tyrosine from Shin et al. 2014** [11]	83,936	124	Null	Sharing	−1.1	0.74	−1.5	0.069
			Null	Causal	−4.0	3.10	−1.3	0.100
			Sharing	Causal	−2.9	2.40	−1.2	0.120
**Tyrosine from Kettunen et al. 2016** [37]	75,155	37	Null	Sharing	0.0033	0.00061	5.4	1
			Null	Causal	0.0400	0.00710	5.7	1
			Sharing	Causal	0.0370	0.00650	5.7	1
**Tyrosine from Locke et al. 2019** [38]	260,603	150	Null	Sharing	−0.4	0.49	−0.82	0.210
			Null	Causal	−4.1	2.20	−1.80	0.035
			Sharing	Causal	−3.7	1.80	−2.00	0.020

CAUSE, causal analysis using summary effect; ELPD, expected log pointwise posterior density; SE, standard error. * Model 1 and Model 2 refer to the models being compared (null, sharing, or causal). ** Model fit is measured by Δ Expected Log Pointwise Posterior Density (ELPD); Negative values indicate that model 2 is a better fit.

## Data Availability

The EPIC-Potsdam datasets analyzed during the current study are not publicly available due to data protection regulations. In accordance with German Federal and State data protection regulations, epidemiological data analyses of EPIC-Potsdam may be initiated upon an informal inquiry addressed to the secretariate of the Human Study Center (Office.HSZ@dife.de). Each request will then have to pass a formal process of application and review by the respective PI and a scientific board. Summary-level data for genetic associations with type 2 diabetes and fasting insulin have been contributed by the DIAGRAM consortium (http://diagram-consortium.org/downloads.html; accessed 08.08.2019) and the MAGIC consortium (https://www.magicinvestigators.org/; accessed 01.04.2020). GWAS-Summary data from Shin et al., 2014, and data on BMI, waist circumference, fasting insulin and (fasting) glucose were accessed via the MR-Base platform and TwoSampleMR R package [24]. The authors thank all investigators for sharing these data.

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
