# Peer review of "Mendelian Randomization Study on Amino Acid Metabolism Suggests Tyrosine as Causal Trait for Type 2 Diabetes"

_nutrients, 2020, doi:10.3390/nu12123890_

Round 1

Reviewer 1 Report

In the manuscript, the authors have tried to reveal causal relationship between branch -chain amino acids and risk of type 2 diabetes (T2DM) using mendelian randomization (MR). The topics, theme, and study design of manuscript are quite interesting, and authors comprehensively summarize their findings. I have concerned minor points to respond.

MINOR:

  1. In materials and methods of 2.1 workflow. It should be better to make a figure to explain flow chart.
  2. In materials and methods of 2.2.3 summary-level data for T2DM, please clearly explain diagnostic criteria of type 2 diabetes.

Reviewer 2 Report

This study aims to evaluate the causal relationships between amino acids (and their ratios) and type 2 diabetes in a mendelian randomization (MR) framework. Reverse causal relationships were also evaluated in a reverse MR analysis. It performed genome-wide association analysis in the EPIC-Potsdam study to identify genetic instruments for some exposures. It also took advantage of existing GWAS summary statistics. There are some overlapped exposures between the two studies. Not all overlapped exposures were used in the same analysis. It is actually a good idea to use different summary statistics for the same exposure as replications. The use of weak genetic instrument is a limitation and it is unclear if the current study has sufficient power. The manuscript is confusing at some places and it will benefit from careful proof-reading. Overall, it is an important study but it needs major revisions to address the following concerns.

  • Abstract Line 31: “However, whether those associations reflect causal relationships or rather preclinical processes of disease development is unclear.” This sentence is confusing. It implicates that preclinical processes of disease development do not represent causal relationships. Why not? The same confusion applies to Line 51-52.
  • The study used amino acids-related exposures from two different studies, the EPIC-Potsdam study and the Shin et al. study. First, please explicitly list the amino acids and their ratios from each study. In line 110, please list all the single amino acids from the Shin et al. study. In line 151, please list all the single amino acids and ratios from the EPIC-Potsdam study. Second, please explain why these amino acids and ratios were selected? I am especially curious about why not more single amino acids were used from the EPIC-Potsdam study? These could serve as replications.
  • Supplementary Table S1, Tables 2 and 3, why are there only 9 exposures? There should be ten (eight amino acid ratios and two single amino acids, according to line 151).  
  • Grammar error. Line 67 “Although, previous MR studies used large data resources, they focused on single amino acids only and did not account for their interrelations within human metabolism.”.
  • Line 74, please add a brief description of “EPIC-Potsdam”.
  • Line 81, “Third, we evaluated the genetic instruments for enriched pathways facilitating biological interpretation” should be “Third, we evaluated the pathway enrichment of the genetic instruments to facilitate biological interpretation”.
  • Line 84, “Forth” should be “Fourth”.
  • Line 85, “To complement our results achieved within EPIC-Potsdam data and to retain higher power to generate genetic instruments for single amino acids” should be “To complement our results achieved with the EPIC-Potsdam data and to gain higher power to generate genetic instruments for single amino acids”.
  • Line 90, it is not clear what “in that context” refers to.
  • Line 114, “We selected this data source, as all investigated amino acid traits from EPIC-Potsdam are available.” But why didn’t you include these single amino acid traits from EPIC-Potsdam?
  • Line 198, the use of IV2 in “the IV2 assumption” is not necessary. There is a specific order for IV1, IV2, and IV3.
  • Line 206, “retain” should be “gain”.
  • Line 211, please describe how genetic instruments were selected for multivariable MR analysis.
  • Line 221, “tree” should be “three”. Also, please briefly explain why two additional datasets were used here for tyrosine. Why weren’t they used in the typical MR analysis (IVW)?
  • Line 300, the descriptions here are incomplete. The results showed that there are inverse associations with Fasting blood glucose adjusted for BMI (p = 0.008) and Blood glucose (p = 0.004).
  • Line 323, “independent whether” should be “independent of whether”.
  • Line 333, “result into” should be “result in”.
  • Line 333-337, the sentence is confusingly long. Please re-write.
  • Line 364-267, the sentence is confusingly long. Please re-write.
  • As noted in the manuscript, almost all genetic instruments are weak (F statistic < 10). It is unclear if the current study has enough statistical power to detect specific associations. It will be more informative and helpful for result interpretation if the authors can provide a supplementary table of statistical power. For example, assuming an effect size of 1.2, what is the power for all pairs of amino acids and type 2 diabetes?

Round 2

Reviewer 2 Report

The revision has significantly improved the quality of the manuscript. One additional analysis is required to confirm the results. There are also two minor edits required. 

Round1: Abstract Line 31: “However, whether those associations reflect causal relationships or rather preclinical processes of disease development is unclear.” This sentence is confusing. It implicates that preclinical processes of disease development do not represent causal relationships. Why not? The same confusion applies to Line 51-52.

  • Response: Thank you for pointing this. We agree that this phrasing might be misleading and changed the wording: “However, whether those associations reflect causal relationships or are rather driven by early processes of disease development is unclear.” (line 31)
  • Round2: Please revise Line 51-52 in a similar way.

Round2: There is a typo in Fig. S3B. “tarits” should be “traits”.

Round1: Line 221, “tree” should be “three”. Also, please briefly explain why two additional datasets were used here for tyrosine. Why weren’t they used in the typical MR analysis (IVW)?

  • Response: We corrected the mistake. We included additional datasets for the Cause-method to use all public available data sources for tyrosine measures. We did not include those datasets for the multivariable MR, as not all of the amino acids, we used before, are available. E.g. Kettunen et al. 2016 does not include measures of tryptophan, methionine and serine. The same is true for Locke et al. 2019
  • Round2: But why weren’t the two datasets used to test the effect of tyrosine on Type 2 diabetes in the univariable MR analysis? Currently, the authors found that tyrosine is negative association with T2D using the Shin et al dataset. It will be more re-assuring if they can replicate the results with the Kettunen et al. 2016 and Locke et al. 2019 datasets. Even if the results are not statistically significant, as long as that they in the same direction, it will bring more confidence in this association. Please perform this analysis and report the results.
